# An Early and Sustained Inflammatory State Induces Muscle Changes and Establishes Obesogenic Characteristics in Wistar Rats Exposed to the MSG-Induced Obesity Model

**DOI:** 10.3390/ijms24054730

**Published:** 2023-03-01

**Authors:** Matheus Felipe Zazula, Diego Francis Saraiva, João Lucas Theodoro, Mônica Maciel, Eliel Vieira dos Santos Sepulveda, Bárbara Zanardini de Andrade, Mariana Laís Boaretto, Jhyslayne Ignácia Hoff Nunes Maciel, Gabriela Alves Bronczek, Gabriela Moreira Soares, Sara Cristina Sagae Schneider, Gladson Ricardo Flor Bertolini, Márcia Miranda Torrejais, Lucinéia Fátima Chasko Ribeiro, Luiz Claudio Fernandes, Katya Naliwaiko

**Affiliations:** 1Laboratório de Plasticidade Morfofuncional, Departamento de Biologia Celular, Setor de Ciências Biológicas, Universidade Federal do Paraná, Curitiba 81530-000, Paraná, Brazil; 2Laboratório de Biologia Estrutural e Funcional, Centro de Ciências Biológicas e da Saúde, Universidade Estadual do Oeste do Paraná, Cascavel 85819-110, Paraná, Brazil; 3Laboratório de Estudo de Lesões e Recursos Fisioterapêuticos, Centro de Ciências Biológicas e da Saúde, Universidade Estadual do Oeste do Paraná, Cascavel 85819-110, Paraná, Brazil; 4Centro de Pesquisa em Obesidade e Comorbidades, Departamento de Biologia Estrutural e Funcional, Universidade Estadual de Campinas, Campinas 13083-970, São Paulo, Brazil; 5Laboratório de Fisiologia Endócrina e Metabolismo, Centro de Ciências Biológicas e da Saúde, Universidade Estadual do Oeste do Paraná, Cascavel 85819-110, Paraná, Brazil; 6Laboratório Experimental de Morfologia, Centro de Ciências Médicas e Farmacêuticas, Universidade Estadual do Oeste do Paraná, Cascavel 85819-110, Paraná, Brazil; 7Laboratório de Metabolismo Celular, Departamento de Fisiologia, Setor de Ciências Biológicas, Universidade Federal do Paraná, Curitiba 81530-000, Paraná, Brazil

**Keywords:** hyperinsulinemia, monosodium glutamate, metabolic syndrome, distress oxidative, pro-inflammatory profile, skeletal muscle fiber types

## Abstract

The model of obesity induced by monosodium glutamate cytotoxicity on the hypothalamic nuclei is widely used in the literature. However, MSG promotes persistent muscle changes and there is a significant lack of studies that seek to elucidate the mechanisms by which damage refractory to reversal is established. This study aimed to investigate the early and chronic effects of MSG induction of obesity upon systemic and muscular parameters of Wistar rats. The animals were exposed to MSG subcutaneously (4 mg·g^−1^ b.w.) or saline (1.25 mg·g^−1^ b.w.) daily from PND01 to PND05 (*n* = 24). Afterwards, in PND15, 12 animals were euthanized to determine the plasma and inflammatory profile and to assess muscle damage. In PND142, the remaining animals were euthanized, and samples for histological and biochemical analyses were obtained. Our results suggest that early exposure to MSG reduced growth, increased adiposity, and inducted hyperinsulinemia and a pro-inflammatory scenario. In adulthood, the following were observed: peripheral insulin resistance, increased fibrosis, oxidative distress, and a reduction in muscle mass, oxidative capacity, and neuromuscular junctions, increased fibrosis, and oxidative distress. Thus, we can conclude that the condition found in adult life and the difficulty restoring in the muscle profile is related to the metabolic damage established early on.

## 1. Introduction

The model of obesity induced by perinatal injections of monosodium glutamate is widely studied and known in the literature [1,2,3]. The main alteration determined in this model is the damage and cell death of neurons in the hypothalamic nuclei, mainly in the arcuate nucleus (ARC), where this neuronal loss impairs the signaling mediated by insulin and affects the energy balance of the organism [3,4,5,6].

Due to the hyperphagic characteristic of the MSG model [7], the excessive consumption of nutrients is associated with the energy imbalance promoted by the hypothalamic lesion. In this model, obesity is associated with the secretion of pro-inflammatory cytokines by adipose tissue, which leads to insulin resistance, stimulating cell damage and impairing metabolic homeostasis in the adipose tissue, liver, pancreas, brain, and muscles [1,2,8,9,10,11,12,13].

It is recognized in the literature that insulin sensitivity and resistance depend on AMPK-mediated signaling pathways, where the main effect of this pathway is the increased GLUT4 translocation in the membranes of insulin-dependent tissues [8,14,15]. As a consequence of this activation, there is a reduction in the phosphorylation rate of the mTOR protein [16,17]. As insulin sensitivity is reduced, especially in skeletal muscle, mTOR signaling which has been implicated in insulin resistance and obesity pathogenesis contributes to the development of the inflammatory process by stimulating the activation of the NfᴋB pathway [1,17,18,19,20].

The reduction in the body growth of animals exposed to the MSG model has been evidenced by several authors [1,9,21,22,23]. This reduction can be identified immediately after the induction period, depending on the concentration and frequency of injections, and is also confirmed in adulthood [22,24]. In addition, the changes in growth caused by this model are not reversible when considering muscle tissue, which suggests that metabolic impairment may be established earlier than has been described in the literature and may resemble models of metabolic programming that affect muscle development and maturation [25,26,27].

Although the endocrine, metabolic, and autonomic aspects of obesity induced by MSG have been extensively studied and described for adult animals, the early effects of MSG exposure and the establishment of muscle changes are less understood and have been little explored. Thus, the present study aimed to identify whether the changes found in adulthood were established early by exposure to MSG.

## 2. Results

### 2.1. Murinometric Profile

#### 2.1.1. Lactation Period

To access the effect of MSG injections on the animals’ developmental delay, the pups were weighed and measured every 2 days. Evaluating the body weight of these offspring, we could observe that from PND07 onwards, MSG animals showed reduced weight gain when compared to CTL (*p* < 0.0001; Figure 1A), and this difference persisted until euthanasia (PND15). Likewise, MSG animals showed less gain in nasoanal length from PND07 when compared to the CTL group (*p* < 0.0001; Figure 1B). However, when evaluating the Lee index, it was only possible to observe lower values in MSG animals in PND09 (*p* = 0.002; Figure 1C). However, when we observed the total gain during the period, through the calculation of the area under the curve, we could observe that in terms of body weight (*p* < 0.0001; Figure 1A’), nasoanal length (*p* < 0.0001; Figure 1B’), and in the Lee index (*p* = 0.0223; Figure 1C’), MSG animals showed reduced development.

#### 2.1.2. Post-Weaning Period

To assess whether the effect of MSG injections would persist into adulthood and whether it participates in the onset of metabolic syndrome and obesity, the animals were weighed, and their food intake was measured once a week from weaning until euthanasia (day 142 of age). The percentage of body weight gain (*p* < 0.0001; Figure 2A) in MSG animals was minor when compared to CTL from week 8 (*p* = 0.0134). Interestingly, this difference was accentuated from week 10, the period where puberty started (*p* < 0.001) and worsened from week 14 with the onset of adulthood (*p* < 0.0001). As expected, the MSG effect on general food consumption was higher in MSG animals when compared to CTL animals (*p* = 0.0019; Figure 2B). Furthermore, to confirm those findings, the area under the curve of total body weight gain and total food consumption was calculated. In both situations, differences between the groups were found. Thus, the total weight gain was lower in MSG animals when compared to CTL (*p* = 0.0026; Figure 2A’), while the total food intake was higher in MSG animals when compared to CTL (*p* = 0.0018; Figure 2B’). 

### 2.2. Intraperitoneal Glucose Tolerance Test (ipGTT) and Insulin Measurement

A glucose tolerance test was performed to assess whether alteration of glucose sensibility and metabolism had been established at 135 days of life, followed by a measurement of plasma insulin levels. In this sense, when we evaluated the response in the ipGTT test (*p* < 0.0001; Figure 3A), we could observe that even with no difference in baseline blood glucose levels between the groups, the MSG animals had higher blood glucose levels at T15 (*p* = 0.0167) and a smaller decrease in blood glucose at T30 (*p* = 0.0492) when compared to CTL. When assessing insulin levels (*p* < 0.0001; Figure 3B), it was identified that MSG animals had elevated basal insulin concentrations when compared to CTL animals. Furthermore, this elevation of insulin concentrations was maintained throughout the test when compared to CTL, suggesting a picture of persistent hyperinsulinemia. When calculating the area under the curve for the total concentration of glucose and insulin throughout the experiment, it was observed that the MSG animals had higher glycemia (*p* = 0.0317; Figure 3A’) and this was accompanied by higher plasma insulin (*p* < 0.0001; Figure 3B’).

### 2.3. Inflammatory and Obesogenic Scenario at 15 Days

#### 2.3.1. Corporal Characterization

On day PND15, MSG animals presented lower weight (*p* < 0.0001) and nasoanal length (*p* = 0.0002) when compared to CTL. However, when calculating the Lee index, there was no difference between the groups (*p* = 0.6851) Figure 4.

#### 2.3.2. Plasmatic Profile

When we evaluated the plasma of MSG and CTL animals, no differences were identified for glucose (*p* = 0.312), total cholesterol (*p* = 0.9248), LDL (*p* = 0.6471), VLDL (*p* = 0.9951), triacylglycerols (*p* = 0.9951), and HDL (*p* = 0.3501). When calculating the dyslipidemia predictors, no change was identified in the lipid ratio (*p* = 0.4318), in the Castelli index 1 (*p* = 0.9485), and in the Castelli index 2 (*p* = 0.9485). Similar results were identified for muscle damage markers, where lactate (*p* = 0.7789) and creatine kinase (*p* = 0.8872) levels were not different in MSG animals when compared to CTL, Figure 5.

Interestingly, when measuring insulin levels and calculating HOMA-IR, it was observed that MSG animals showed an increase when compared to CTL (*p* < 0.0001, in both). Meanwhile, the calculation of QUICKI (*p* < 0.0001) showed a reduction in MSG animals when compared to CTL. However, MSG animals showed increased concentrations of TNFα (*p* = 0.0284), IL-06 (*p* < 0.0001), and IL-10 (*p* = 0.0053), Figure 6.

#### 2.3.3. Skeletal Muscle Antioxidant System and Oxidative Damage

When evaluating the antioxidant system and oxidative damage of the skeletal muscle pool, it was again identified that there were no differences between the MSG and CTL groups in the activity of the enzymes superoxide dismutase (*p* = 0.5249), catalase (*p* = 0.5198), and total cholinesterase (*p* = 0.3159), as well as in the levels of soluble proteins (*p* = 0.8843), lipid peroxides (*p* = 0.5253), and non-protein thiols (*p* = 0.9972), Figure 7.

### 2.4. Obesity and Muscle Damage at 142 Days

#### 2.4.1. Corporal Characterization

The analysis of the animals in the PND142 showed that the MSG animals had a lower weight (*p* < 0.0001) and nasoanal length (*p* = 0.0002) when compared to the CTL animals. However, when compared to the CTL animals, the MSG animals had a higher Lee index (*p* = 0.0153) and higher adiposity (*p* < 0.0001), Figure 8.

#### 2.4.2. Plasmatic Profile

When the plasma profile of the animals was evaluated, the MSG animals showed an increase in glucose (*p* = 0.0011), total cholesterol (*p* < 0.0001), LDL (*p* < 0.00,01), and VLDL (*p* = 0.0052) cholesterol fractions, as well as in total triacylglycerols levels (*p* = 0.0052). There was no difference in HDL between the groups (*p* = 0.0657). When performing the calculation of dyslipidemia predictors, an increase in the lipid ratio was identified (*p* = 0.0115), in the Castelli index 1 (*p* = 0.0001), and the Castelli index 2 (*p* = 0.0061). Likewise, when evaluating some muscle damage markers, an increase in lactate levels (*p* = 0.0008) and a decrease in creatine kinase concentrations (*p* < 0.0001) were identified in the MSG animals when compared with the CTL animals, Figure 9. 

Interestingly, when measuring the insulin levels and calculating HOMA-IR, it was observed that the MSG animals showed an increase when compared to the CTL animals (*p* < 0.0001, in both). Meanwhile, the calculation of QUICKI (*p* < 0.0001) showed a reduction in MSG animals when compared to the CTL animals, Figure 10.

#### 2.4.3. Skeletal Muscle Structure

When we evaluated the macroscopic characteristics of the muscle, the MSG animals had lower EDL muscle weight (*p* < 0.0001) and shorter SOL (*p* = 0.0011) and EDL muscle length (*p* < 0.0001) when compared to the CTL animals. When evaluating the muscular structure of the EDL and SOL, it was identified that the MSG animals had a higher density of fibers per mm^2^ in both muscles (*p* = 0.0022; *p* < 0.0001, respectively), when compared to the CTL animals, accompanied by a reduction in the cross-sectional area of muscle fibers, observed in both muscles of the MSG animals (*p* = 0.0020; *p* < 0.0001, respectively). In addition, we found a reduction in the largest (*p* = 0.0003; *p* < 0.0001, respectively) and smallest (*p* = 0.0006; *p* < 0.0001, respectively) diameters in both muscles of the MSG animals when compared to the CTL animals. It was also possible to identify that the SOL of the MSG animals showed a reduction in the diameter ratio, an important predictor of muscle fiber rounding (*p* = 0.0178), Table 1.

Another feature evaluated was the distribution of capillaries and nuclei in the cells of both muscles. MSG animals showed a greater distribution of capillaries in EDL when compared to CTL (*p* < 0.0001), while the distribution of capillaries was reduced in SOL (*p* < 0.0001). In the distribution of nuclei, the MSG animals showed lower values in SOL (*p* < 0.0001) when compared to the CTL animals. However, when we evaluated the presence of nuclei in a central position in the muscle fibers, we could observe that for both the EDL and SOL muscles, the MSG animals showed an increase comparable to the CTL animals (*p* < 0.0001, in both). In the case of the myonuclear domain, there was a reduction in MSG animals compared to the CTL animals for EDL only (*p* = 0.0081), Table 1.

When evaluating the distribution of connective tissue in the EDL and SOL muscles, it was found that in MSG animals there was an increase in total connective tissue in both muscles when compared to the CTL animals (*p* < 0.0001; *p* = 0.0008, respectively). In addition, in the MSG animals higher values of connective tissue in the epimysium (*p* < 0.0001, in both) and perimysium (*p* < 0.0001, in both) in both muscles were shown. However, endomysium thickening was identified only in the EDL of MSG animals (*p* = 0.0004). The evaluation of the type of collagen in each of the muscles revealed that in the EDL, the MSG animals showed a reduction in type I collagen (*p* = 0.0014) and an increase in type III collagen (*p* = 0.00014), while in the SOL, only type III collagen reduction (*p* = 0.0078) could be identified in MSG animals when compared to the CTL animals, Table 1. 

#### 2.4.4. Fiber Types Profile and Neuromuscular Junction Structure

When analyzing the prevalence of each type of f, the MSG animals showed a reduction in the proportion of type I fibers in EDL and SOL (*p* < 0.0001, in both) and a reduction in the cross-sectional area of type I fibers in EDL and SOL (*p* < 0.0001, in both) when compared to the CTL animals. In SOL, MSG animals showed an increase in the proportion of IIA-type fibers (*p* < 0.0001), and in both muscles, EDL and SOL, there was a reduction in the cross-sectional area of IIA-type fibers (*p* = 0.0093; *p* < 0.0001, respectively) of the MSG animals when compared to the CTL. Furthermore, it was identified that the MSG animals showed a reduction in the cross-sectional area of the neuromuscular junctions both in the EDL (*p* = 0.0167) and in the SOL (*p* < 0.0001) when compared to the CTL animals. The MSG animals showed a reduction in the major (*p* < 0.0001) and minor (*p* < 0.0001) diameters in the SOL muscle junctions when compared to the CTL animals. Therefore, when evaluating the ratio between the largest and smallest diameters, a predictor of damage to the structure, it was observed that the MSG animals had a lower ratio (*p* = 0.0102) when compared to the CTL animals. Finally, when analyzing the antioxidant system and oxidative damage in EDL and SOL, it was identified that catalase activity was reduced in EDL (*p* = 0.0101) and increased in SOL (*p* < 0.0001) in MSG animals compared to the CTL animals. However, in both muscles, the MSG animals showed an increase in the concentration of proteins (*p* = 0.0268, in both), of lipid peroxides (*p* = 0.0405; *p* < 0.0001, respectively), and a reduction in the concentration of non-protein thiols (*p* < 0.0001; *p* = 0.0003, respectively), when compared to the CTL animals. Finally, when we evaluated the total cholinesterase activity in both muscles, the MSG animals showed a decrease in activity when compared to the CTL animals (*p* = 0.0001; *p* = 0.0003, respectively), Table 1.

### 2.5. Multivariate Analysis

When the interaction between the variables was evaluated, it was observed that the MSG animals already had body impairment characteristics of the model in the PDN15 (F_1,10_ = 2.7748, R^2^ = 0.2172, *p* = 0.0021, Figure 11A). These characteristics are due to the delay in body development (F_1,10_ = 13.4489, R^2^ = 0.5735, *p* = 0.0027, Figure 11B) and the inflammatory profile established in the animals, associated with the state of hyperinsulinemia (F_1,10_ = 28.0549, R^2^ = 0.7372, *p* = 0.0025, Figure 11E). Despite these findings, changes in plasma (F_1,10_ = 0.2948, R^2^ = 0.0286, *p* = 0.9722, Figure 11C) or in the muscle antioxidant system (F_1,10_ = 0.3958, R^2^ = 0.0381, *p* = 0.2875, Figure 11D), which are commonly described as fundamental factors for establishing of the condition in adult animals, were not identified at this age. We also observed that the alterations observed in the young animals intensified in adulthood, producing the body impairment characteristic of MSG induction (F_1,14_ = 18.3229, R^2^ = 0.5668, *p* = 0.0002, Figure 12A). These model characteristics are due to delayed body development, reduced muscle mass, and fat accumulation (F_1,14_ = 24.7399, R^2^ = 0.6386, *p* = 0.0002, Figure 12B). In this sense, it is possible to identify the establishment of the metabolic syndrome in these animals (F_1,14_ = 21.0869, R^2^ = 0.6009, *p* = 0.0002, Figure 12C). These factors are fundamental for the impairment identified in the muscle structure, such as the reduction in fiber size, alteration in the distribution of nuclei and capillaries, and thickening of the connective envelopes (F_1,14_ = 20.9169, R^2^ = 0.5991, *p* < 0.0001, Figure 12D). It was also possible to observe a reduction in the oxidative capacity of muscle fibers (F_1,14_ = 19.5689, R^2^ = 0.5629, *p* = 0.0001, Figure 12E) and a reduction in neuromuscular junctions (F_1,14_ = 12.8698, R^2^ = 0.4789, *p* = 0.0002, Figure 12F), in addition to the accumulation of oxidative damage markers accompanied by impairment of the muscular antioxidant system (F_1,14_ = 12.8419, R^2^ = 0.4784, *p* = 0.0002, Figure 12G).

## 3. Discussion

The literature has reported the effects of MSG as an inducer of obesity, where the main object of study is adult animals with obesity already installed. Here, we present a new study proposal, where the main objective was to investigate whether exposure to MSG in the first days of postnatal life could produce early metabolic changes. In this study, instead of evaluating only the conditions of the animals in the adult phase, we sought to identify the characteristics of the animals 10 days after the end of the monosodium glutamate injections. The main results obtained agree with the results established in the literature for the MSG-obesity model; however, significant alterations in the inflammatory and insulin profile were identified, early in the installation of obesity parameters. These results suggest a slightly different scenario from that classically found for this model, in which damage to the hypothalamic nuclei may be associated with early identified pro-inflammatory disorders and hyperinsulinemia. Thus, it is likely that the muscle changes induced by the model are due not only to the chronicity of the metabolic condition in adulthood but also to this metabolic pattern established early on.

The induction of obesity by MSG causes cytotoxic damage to the hypothalamic nuclei, which induces significant changes in the development of animals, mainly due to cell loss in the GH-secreting hypothalamic nuclei [28], as has been well described in the literature [7] and previously identified in works by our research group [9,21,29,30]. The relationship between reduced body growth and reduced muscle growth has also been extensively explored [1,22,28], with a consensus in the literature that MSG cytotoxicity also results in a model of short stature due to hormonal insufficiency that leads to low growth [22,31]. In this sense, the reduction in body weight gain accompanied by a lower nasoanal length in the MSG animals suggests that from the second day after the end of the injections (PND07) such changes are being established, corroborating that the effects of MSG reach different tissues, since the start of the exhibition.

It is known that GH participates in the close relationship between factors that repress the development and differentiation of muscle fibers, such as myostatin, and that in the MSG-induced obesity model, GH-secreting hypothalamic nuclei are affected, altering GH secretion. Thus, it is possible that in this model this injury results in the attenuation of the feedback mechanisms that repress myostatin activity, producing a reduction in the size of muscle fibers and altering the proportion of fiber types during the final process of development [32,33]. Here, the data obtained demonstrate that in the presence of MSG, muscle fibers are smaller, suggesting the participation of regulation mediated by GH-myostatin, in the reduction of muscle fiber size. Furthermore, such results may be due to an imbalance in the secretion of growth factors, due to muscle damage induced by the obesity model. The model may promote the reduction of growth factors, such as the fibroblast growth factor, required by the muscle for proliferation, as well as for the growth and differentiation of mesenchymal cells during development [8,34,35].

It has also been reported that the metabolic changes associated with the model may originate from lesions that occur in several central structures of the paraventricular region of the hypothalamus, where the arcuate and ventromedial nuclei are the most affected. It is believed that about 80 to 90% of the control of food consumption, energy expenditure, and glucose homeostasis is due to the neuronal activity of these nuclei [12,14,36]. Dysfunction of these structures promotes an imbalance i metabolic pathways, causing the increase in plasma lipid concentrations and their incorporation into adipose tissue, as found in the adult animals of this work [1,12]. Dysfunction of these structures promotes an imbalance of metabolic pathways, causing an increase in plasma lipid concentrations and their incorporation into adipose tissue, as found in the adult animals of this work [37].

By evaluating the levels of insulin and plasmatic cytokines, we could see that through a reduction in glucose sensibility, it is first possible that signaling of peripheral insulin resistance, accompanied by a pro-inflammatory profile, evidenced by increased concentrations of IL-6 and TNFα, is already identifiable in PND15. The increase in IL-6 associated with the MSG model has an important effect on muscle development, as it reduces IGF-1 secretion and muscle sensitivity to insulin, negatively modulating the differentiation, and growth of muscle fibers [38,39,40]. In models of dietary obesity, increased IL-6 secretion has also been associated with reduced muscle mass [41,42]; however, in models that use MSG exposure, there is a recurrent reduction in the secretion of this cytokine in animals. The effect of MSG was also evaluated in adulthood, which supports the idea that there may be early metabolic programming in the active phase of obesity [43,44]. Finally, we found an increase in plasma TNFα secretion associated with this scenario, which may be related to the lower availability of MyoD for the paracrine effect, causing a reduction in the differentiation of myoblasts into myocytes, in addition to a reduction in the fusion of myotubes, which is implicated in the muscular alterations identified in the study. In addition, the increase in TNFα is related to reduced insulin sensitivity, increased muscle catabolism, sarcomere ubiquitination, and NADPH oxidation, which together may negatively modulate muscle development and differentiation [38,39,40,41].

The characteristics related to the number, position, and structure of the myonuclei of MSG-obese animals in the present study, have been associated with the response mediated by the chronic stress resulting from the established metabolic syndrome [45,46], which may be indicative of damage caused by the incomplete state of differentiation muscle, resulting from the early inflammatory process. Furthermore, this set of changes found in the proposed obesity model is essential to induce the phenotypic transition of muscle fibers [47,48,49]. The condition of insulin resistance promoted by the inflammatory process, which becomes chronic due to the dyslipidemic profile, is a determining factor for the reduction of muscle oxidative capacity, especially when associated with the characteristics of reduced size and the number of types I and IIA fibers and the increase in type I fibers and IIB fibers [49,50,51].

The establishment of the early hyperinsulinemic condition, found in the PND15 of MSG-obese animals, may indicate the anticipation of the dynamic phase of obesity, where the induction of increased glucose uptake by insulin-responsive tissues seems to occur. However, maintenance of this condition in PND142, where obesity was chronic due to persistent damage, reinforces the establishment of peripheral reduction in glucose sensibility, as observed in MSG animals [52,53,54]. During the worsening of obesity, damage resulting from MSG-induced hepatotoxicity is common, which causes an increase in the generation of reactive oxygen species and feeds back the inflammatory process [15,55,56]. In addition, the inflammatory process, mainly mediated by the increase in TNFα, negatively and significantly modulates the rate of lipolysis, favouring adipose tissue hypertrophy and an increase in fat panicles [12,31,57].

Something intriguing in the MSG cytotoxicity model of obesity induction is the difficulty in applying treatment protocols that restore the physiological state of these animals, after the establishment of obesity. In a resistance exercise model, obese MSG animals submitted to the training protocol showed partial reversal of the obesogenic parameters, but even though they were significant, the reduction in lipemia and adipose panicles did not return to the values found in the control animals [15]. Likewise, after applying a swimming model, the reduction in adiposity and insulin secretion did not return to physiological patterns, and changes in intestinal structure were still persistent [58]. Previous data from our research group show that whole-body vibration training was not able to completely repair the soleus [29,59], extensor digitorum longus [30], tibialis anterior [21], and diaphragm [60] muscles despite promoting anti-obesogenic effects. Furthermore, it did not restore the physiological levels, biochemical, and structural parameters of the liver, adipose tissue, and plasma [9]. Finally, models with leucine [61] and taurine [62] supplementation, as well as herbal treatment [63], showed a partial reduction in body adiposity and food intake, accompanied by improvement in glucose metabolism and insulin sensitivity and cardiovascular effects.

Considering all the initial characteristics of the neurotoxic effect of MSG, associated with the hyperinsulinemic and pro-inflammatory condition in the dynamic phase of obesity induction, we can correlate these developmental alterations with the muscular characteristics found. Thus, as obesity becomes chronic and metabolic syndrome sets in, it is reasonable that most studies find similar results in obese adult animals under the influence of MSG. Although there were several forms of administration, the doses and periods proposed and evaluated, the analyses have the observation of animals at a certain moment in common, when obesity is already well established. Despite this limitation, it is common to find studies that manage to partially repair the damage caused, a fact that reinforces our hypothesis that early damage is established and prevents the correct development of animals, suggesting that exposure to MSG, in the perinatal phase, is capable of inducing some metabolic programming that becomes worse over time. Thus, although the MSG model is remarkably effective in inducing obesity, limitations arising from the proposed study setting may underestimate the systemic effects of MSG and the possible effects of treatment protocols. Finally, considering the role of muscle tissue in metabolic regulation, obtaining results that support muscle restructuring may represent an important strategy for improving metabolic conditions, even in the absence of the reversal of body parameters of obesity.

## 4. Materials and Methods

### 4.1. Ethical Approval

All trials in this study were conducted following national and international recommendations and legislation [64] and with the approval of the University Animal Care Committee (protocol # 08/18).

### 4.2. Animals and Experimental Design

From postnatal day (PND) 01 to PND05, male Wistar rats (*n* = 24) received daily subcutaneous injections of MSG solution in the dorsocervical region (4 mg·g^−1^ body weight, MSG group) or equimolar saline solution (1.25 mg g^−1^ body weight, control group—CTL) [3,9]. Every two days, the animals were weighed and their nasoanal length was measured, until the 14th day of life. In PND15, 12 animals were euthanized (*n* = 6 per group) to assess the establishment of molecular damage. After weaning (PND21), food consumption and the evolution of body weight were monitored weekly. All of the animals were housed in standard cages at a constant temperature (22 ± 1 °C), on a 12 h light-dark cycle, and had ad libitum access to water and standard laboratory chow (BioBase^®^, Santa Catarina, Brazil).

### 4.3. Intraperitoneal Glucose Tolerance Test (ipGTT) and Insulin Dosage

The ipGTT was performed after eight hours of fasting and consisted of a small cut in the tail of the animals followed by the collection of blood samples to measure glucose with the aid of an Accu Chek glucometer (Roche Diabetes Care Brasil LTDA, São Paulo, Brazil). Blood was collected in the fasted state (time 0) and 15, 30, 60, and 90 min after IP injection of a glucose overload (2 g·kg^−^^1^ of body weight). Additional blood samples were collected with heparinized glass capillaries and then centrifuged at 4 °C and 12,000× *g* for 10 min. The supernatant was stored in a freezer at −80 °C for later measurement of insulin by radioimmunoassay.

### 4.4. Euthanasia and Material Collection

In PND15, the Lee index (∛bodyweight / nasalanal length × 1000) was calculated. The animals were then desensitized in a carbon dioxide chamber and then euthanized by decapitation [7]. Blood was collected in heparinized tubes and centrifuged at 4 °C, at 12,000× *g* for 10 min to measure the plasma biochemical and inflammatory profile. The abdominal wall and pelvic limb muscles were collected (approximately 0.2 g) and intended for the analysis of oxidative damage markers.

In PND142, the Lee index (∛bodyweight/nasalanal length × 1000) was calculated. The animals were then desensitized in a carbon dioxide chamber and then euthanized by decapitation [7]. Retroperitoneal, perigonadal, and brown fats were removed, weighed, normalized to g·100 g^−1^ of body weight, and used to calculate body adiposity [2]. Blood was collected in heparinized tubes and centrifuged at 4 °C, at 12,000× *g* for 10 min to measure the plasma biochemical profile. The extensor digitorum longus (EDL) and soleus muscle (SOL) were dissected, collected, weighed, measured, and destined for biochemical and morphological analysis.

### 4.5. Skeletal Muscle Structure Analysis

The muscle was sectioned in the middle region of the muscle belly, and the proximal fragments of the right antimere were fixed in metacarn and stored in 70% alcohol. Subsequently, they were submitted to the histological procedure with dehydration in an increasing series of alcohol, diaphanization in N-butyl alcohol, and inclusion and embedding in histological paraffin, after which they were cut transversely at 5 µm with the aid of a microtome. For the study of muscle fibers, the sections were stained with hematoxylin–eosin (HE), morphologically analyzed under a light microscope, and 10 visual fields of interest were photographed at 400× magnification. In the images obtained, the cross-sectional area of the fiber and cores, fiber density, number, and position of nuclei were analyzed.

The distal fragments of the right antimere were used for histoenzymological analysis, which analyzes the oxidative and glycolytic metabolism of muscle fibers. For this, immediately after collection, they were covered with neutral talc for tissue preservation and subsequently frozen in liquid nitrogen, conditioned in cryotubes, and stored in a Biofreezer at −80 °C, up to a 7 µm section in a cryostat chamber (LUPETEC CM 2850 Cryostat Microtome) at −20 °C. The sections were submitted to the enzymatic reaction of NADH–TR (nicotinamide adenine dinucleotide—tetrazolium reductase). This analysis quantifies the different types of muscle fibers (I, IIa, and IIb) according to the tone presented in the fibers after the reaction. For each animal, five microscopic fields were randomly chosen at 200× magnification to count and analyze the area size of the different types of fibers.

The proximal fragments of the left antimere were used for the study of JNM, and they were immersed in Karnovisky’s fixative. The muscles were cut longitudinally into small portions with stainless steel blades, and the selected cuts were found in the nonspecific esterase reaction. Subsequently, a morphological analysis of the slides was performed with a light microscope, photomicrographing the visual fields of interest at 200× magnification. The size of the area, the largest diameter, and the smallest diameter of 150 JNM per animal were measured.

The morphological analyzes were performed in the Image ProPlus 6.0 (Media Cybernetics, Inc., Rockeville, MD, USA) program, and in each image the muscle fasciculus was scanned to randomly select ten fibers, thus totaling 120 fibers per animal.

### 4.6. Antioxidant System and Oxidative Damages Analysis

For the evaluation of the antioxidant system, the distal portion of the left antimer of EDL muscle was homogenized with Tris-HCl buffer (0.4 M, pH 7.4) and centrifugated for 20 min at 4 °C and 12,000× *g*. Tissue protein quantification was determined by the Bradford method, using bovine serum albumin as a standard. All of the samples were normalized to 1 mg protein × mL^−^^1^.

The enzymatic activity of the superoxide dismutase (SOD—EC 1.15.1.1) of the muscles was determined by inhibiting the formation of formazan blue by reducing nitrotetrazolium blue (NBT); increasing absorbance by reducing NBT by the superoxide anion was monitored at 560 nm (RS: 182 mM sodium carbonate buffer pH 10.2; 50 µM EDTA; 100 µM NBT; 36.86 mM hydroxylamine sulfate). The values were expressed in U × mg protein^−^^1^ [65].

The enzyme activity of the catalase (CAT—EC 1.11.1.6) of the muscles was determined through the formation of H_2_O and O_2_ from the consumption of H_2_O_2_, the reduction in absorbance by the consumption of H_2_O_2_ was monitored at 240 nm (RS: 50 mM of potassium phosphate buffer pH 7.0; 10 mM H_2_O_2_). The values were expressed in mM of H_2_O_2_ consumed × min^−^^1^ × mg protein^−^^1^ [66].

The lipid peroxidation index (LPO) of the muscles was determined by the generation of complexes between Fe^+2^ and xylenol orange and the formation of a chromophore stabilized by butylated hydroxytoluene. The absorbance by the generation of the chromophore was measured at 560 nm. The values were expressed in nM hydroperoxides × mg protein^−^^1^ [67].

The enzymatic activity of total cholinesterase (ChE—EC 3.1.1.8) was determined by the generation of 2-nitrobenzoate-5-mercaptothiocholine from the interaction of thiocholine and DTNB; the increase in absorbance by the formation of the chromophore was monitored at 405 nm (RS: 487 µM DTNB; 2.25 mM acetylthiocholine iodide). The values were expressed in nM acetylthiocholine hydrolyzed × min^−^^1^ × mg protein^−^^1^ [68].

### 4.7. Statistical Analysis

Data were expressed as mean ± standard deviation and analyzed using descriptive and inferential statistics in the R program version 4.0.3 (45). Data were evaluated for normality (Shapiro–Wilk test). Parametric data were evaluated by the Student’s *t*-test. In the case of non-parametric data, the test used was the Mann–Whitney U test. In the case of data analyzed over time, the ANOVA test of repeated measures with the post-Tukey’s HSD test was used. In all cases, the significance level adopted was 5%.

The data were ordered in response matrices, and in the PND15 the groupings were: general model damage (all data); body pattern; plasma standard; antioxidant system; inflammation. As for PND142, the groups were: general model damage (all data); body pattern; plasma standard; skeletal muscular structure; fiber type profile; structure of neuromuscular junctions; antioxidant system.

## Figures and Tables

**Figure 1 ijms-24-04730-f001:**
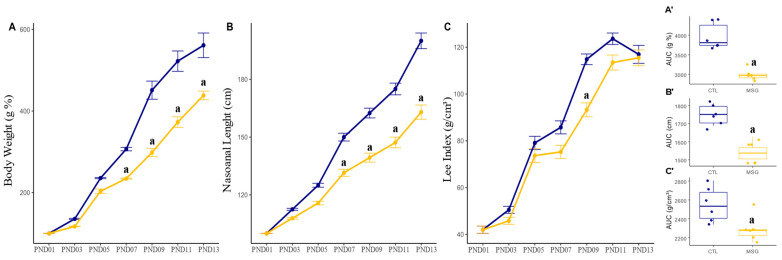
Graphs related to the development of CTL and the MSG animals during the perinatal period (PND01 to PND13). (**A**): Line plot of per cent weight gain (g%); (**B**): line plot of nasoanal length gain (cm); (**C**): Lee’s index gain line plot (g/cm³); (**A’**): AUC of body weight gain; (**B’**): AUC of nasoanal length gain (cm); (**C’**): AUC of the Lee index gain (g/cm³). The CTL is represented in blue and the MSG group is represented in yellow. The letter a represents the difference between the MSG group when compared to the CTL. (**A**–**C**): repeated measurements ANOVA. (**A’**–**C’**): Student’s *t*-test.

**Figure 2 ijms-24-04730-f002:**
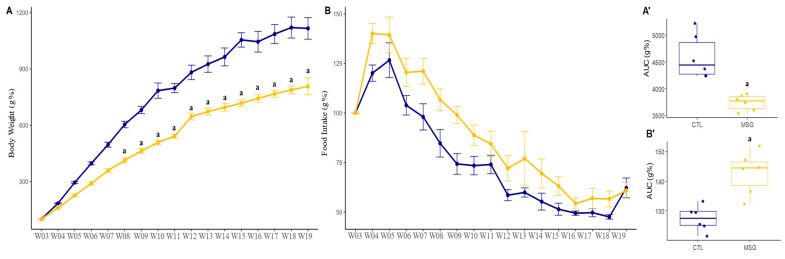
Graphs related to weight gain and feed consumption of CTL and the MSG animals after weaning (PND21, or W03, to PND142, or W19). (**A**): Percent weight gain line graph (g%); (**B**): line plot of food consumption (g%); (**A’**): AUC of body weight gain; (**B’**): AUC of food consumption (g%). The CTL is represented in blue and the MSG group is represented in yellow. The letter a represents the difference between the MSG group when compared to the CTL. (**A**,**B**): repeated measurements ANOVA. (**A’**,**B’**): Student’s *t*-test.

**Figure 3 ijms-24-04730-f003:**
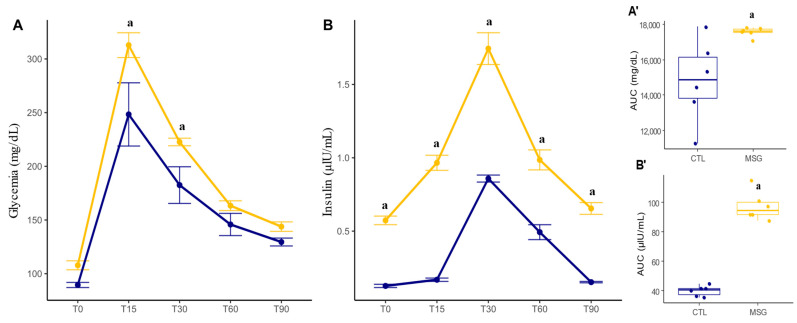
Graphs related to ipGTT and insulin dosage of CTL and MSG animals at 135 days of life. (**A**): Graph of blood glucose during the test (mg/dL); (**B**): insulin level graph (µIU/mL); (**A’**): AUC of blood glucose during the test (mg/dL); (**B’**): AUC of insulin level (µIU/mL). The CTL is represented in blue and the MSG group is represented in yellow. The letter a represents the difference between the MSG group when compared to the CTL. (**A**,**B**): repeated measurements ANOVA. (**A’**,**B’**): Student’s *t*-test.

**Figure 4 ijms-24-04730-f004:**
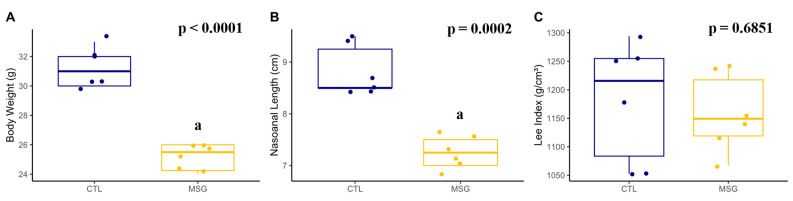
Graphs related to corporal characterization in PND15: (**A**): body weight (g); (**B**): nasoanal length (cm); (**C**): Lee index (g/cm^3^). The CTL is represented in blue and the MSG group is represented in yellow. The letter a represents the difference between the MSG group when compared to the CTL. All data: Student’s *t*-test.

**Figure 5 ijms-24-04730-f005:**
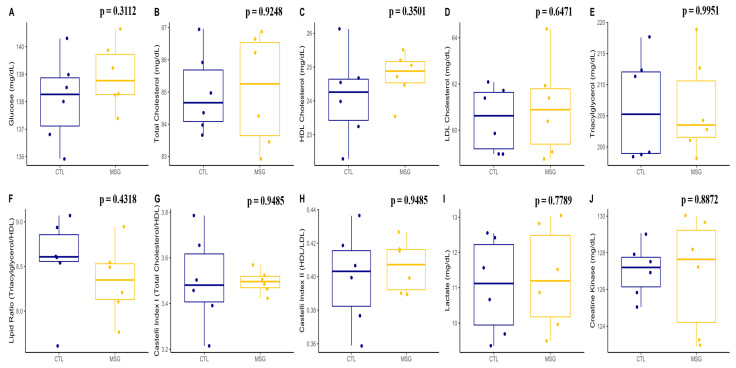
Graphs related to plasmatic profile in PND15: (**A**): glucose (mg/dL); (**B**): total cholesterol (mg/dL); (**C**): HDL cholesterol (mg/dL); (**D**): LDL cholesterol (mg/dL); (**E**): total triacylglycerols (mg/dL); (**F**): lipid ratio (total triacylglycerols/HDL); (**G**): Castelli index I (Total Cholesterol/HDL); (**H**): Castelli index II (HDL/LDL); (**I**): lactate (mg/dL); (**J**): creatine kinase (mg/dL). The CTL is represented in blue and the MSG group is represented in yellow.

**Figure 6 ijms-24-04730-f006:**
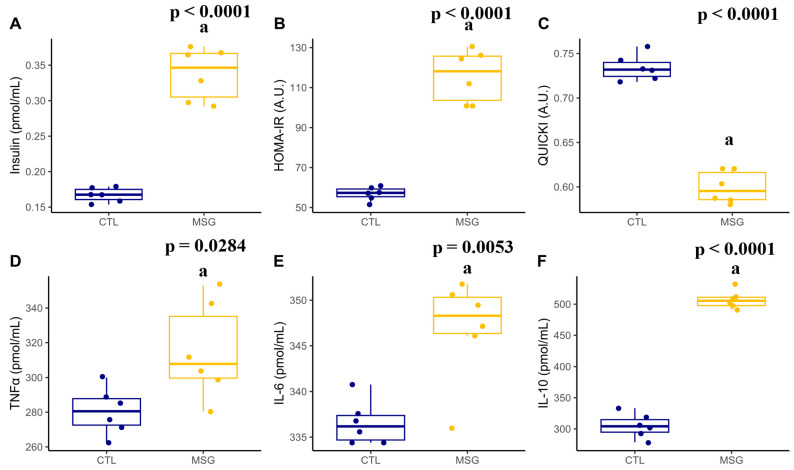
Graphs related to insulin and inflammatory plasmatic profile in PND15: (**A**): insulin (µIU/mL); (**B**): HOMA-R (A.U); (**C**): QUICKI (A.U); (**D**): TNFα (pmol/mL); (**E**): IL-6 (pmol/mL); (**F**): IL-10 (pmol/mL). The CTL is represented in blue and the MSG group is represented in yellow. The letter a represents the difference between the MSG group when compared to the CTL group. All data: Student’s *t*-test.

**Figure 7 ijms-24-04730-f007:**
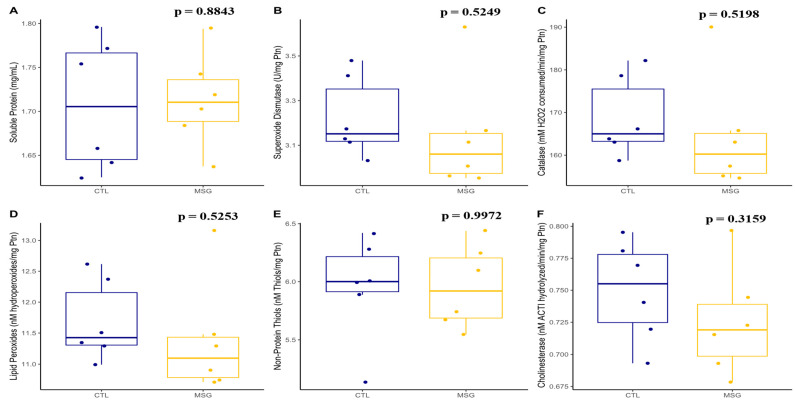
Graphs related to skeletal muscle antioxidant system in PND15: (**A**): soluble proteins (mg/mL); (**B**): superoxide dismutase activity (U/mg protein); (**C**): catalase activity (mM H_2_O_2_ consumed/min/mg protein); (**D**): lipid peroxides (nM hydroperoxides/mg protein); (**E**): non-protein thiols concentration (nM thiols/mg protein); (**F**): cholinesterase activity (nM acetylthiocholine hydrolyzed/min/mg protein). The CTL group is represented in blue and the MSG group is represented in yellow. All data: Student’s *t*-test.

**Figure 8 ijms-24-04730-f008:**
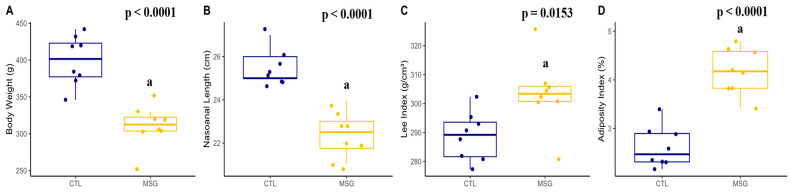
Graphs related to corporal characterization in PND142: (**A**): body weight (g); (**B**): nasoanal length (cm); (**C**): Lee index (g/cm^3^); (**D**): adiposity index (%). The CTL is represented in blue and the MSG group is represented in yellow. The letter a represents the difference between the MSG group when compared to the CTL group. All data: Student’s *t*-test.

**Figure 9 ijms-24-04730-f009:**
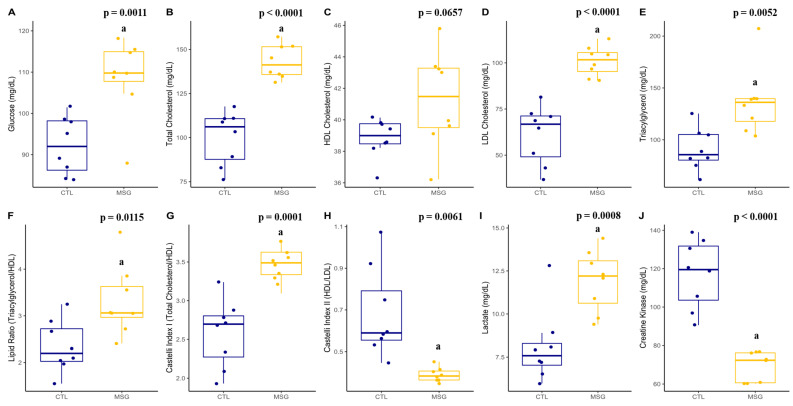
Graphs related to plasmatic profile in PND142: (**A**): glucose (mg/dL); (**B**): total cholesterol (mg/dL); (**C**): HDL cholesterol (mg/dL); (**D**): LDL cholesterol (mg/dL); (**E**): total triacylglycerols (mg/dL); (**F**): lipid ratio (total triacylglycerols/HDL); (**G**): Castelli index I (total cholesterol/HDL); (**H**): Castelli index II (HDL/LDL); (**I**): lactate (mg/dL); (**J**): creatine kinase (mg/dL). The CTL is represented in blue and the MSG group is represented in yellow. The letter a represents the difference between the MSG group when compared to the CTL group. All data: Student’s *t*-test.

**Figure 10 ijms-24-04730-f010:**
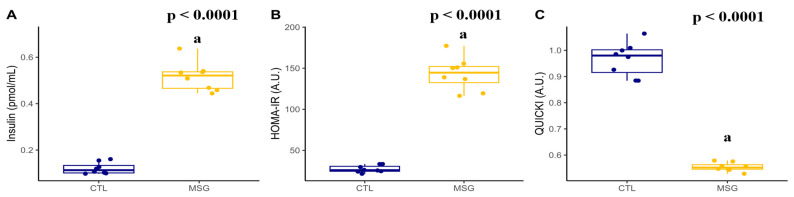
Graphs related to insulin plasmatic profile in PND142: (**A**): insulin (µIU/mL); (**B**): HOMA-R (A.U); (**C**): QUICKI (A.U). The CTL group is represented in blue and the MSG group is represented in yellow. The letter a represents the difference between the MSG group when compared to the CTL group. All data: Student’s *t*-test.

**Figure 11 ijms-24-04730-f011:**
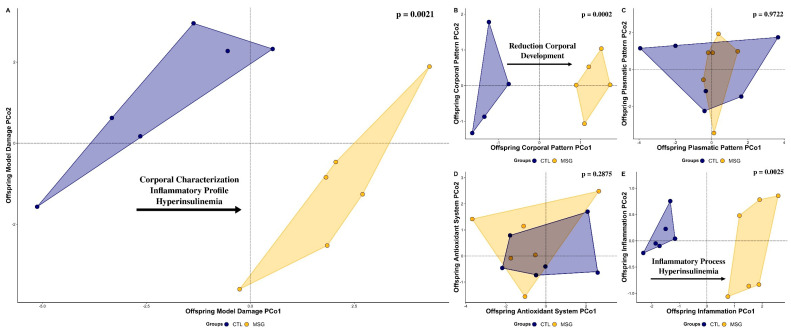
Graphical representation in scatterplot form of the multivariate relationships of the principal coordinates analysis (PCoA) of the animals to the PND15. (**A**): General Model Damage (all data); (**B**): corporal pattern; (**C**): plasmatic pattern; (**D**): antioxidant system; (**E**): inflammation. The CTL group is represented in blue and MSG group is represented in yellow. All data: principal coordinates analysis.

**Figure 12 ijms-24-04730-f012:**
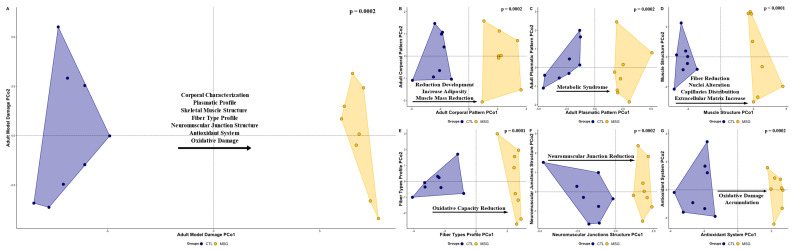
Graphical representation in scatterplot form of the multivariate relationships of the principal coordinates analysis (PCoA) of the animals to the PND142. (**A**): General Model Damage (all data); (**B**): corporal pattern; (**C**): plasmatic pattern; (**D**): skeletal muscle structure; (**E**): fiber types profile; (**F**): neuromuscular junctions structure; (**G**): antioxidant system. The CTL group is represented in blue and MSG group is represented in yellow All data: principal coordinates analysis.

**Table 1 ijms-24-04730-t001:** Skeletal muscle structure, fiber types profile, and neuromuscular junctions structure from CTL and MSG animals to PND142.

Grouping Category	Variable	EDL			SOL		
		CTL	MSG	*p*-Value	CTL	MSG	*p*-Value
Macroscopical Structure	Muscle Weight	0.16 ± 0.01	0.12 ± 0.01 ^a^	<0.0001	0.11 ± 0.01	0.12 ± 0.01	0.1382
	Muscle Length	31.60 ± 1.18	27.55 ± 0.92 ^a^	<0.0001	23.64 ± 1.80	20.38 ± 0.64 ^a^	0.0011
Skeletal Muscle Structure	Fiber Density	574.82 ± 24.01	629.13 ± 32.43 ^a^	0.0022	318.60 ± 27.67	466.44 ± 15.27 ^a^	<0.0001
	Cross-Sectional Area	1743.2 ± 73.4	1595.3 ± 82.2 ^a^	0.0020	3165.9 ± 284.4	2146.2 ± 70.7 ^a^	<0.0001
	Larger Diameter	58.86 ± 7.14	43.37 ± 2.22 ^a^	0.0003	78.51 ± 5.68	57.00 ± 5.59 ^a^	<0.0001
	Smaller Diameter	42.37 ± 5.76	31.31 ± 2.60 ^a^	0.0006	46.77 ± 2.98	36.99 ± 3.83 ^a^	<0.0001
	Diameter Ratio	1.39 ± 0.03	1.39 ± 0.05	0.9332	1.68 ± 0.12	1.54 ± 0.05 ^a^	0.0178
	Capillaries/Fibers	1.50 ± 0.14	3.03 ± 0.41 ^a^	<0.0001	3.28 ± 0.24	2.05 ± 0.02 ^a^	<0.0001
	Nuclei/Fibers	2.12 ± 0.37	1.88 ± 0.19	0.1891	2.72 ± 0.09	1.85 ± 0.08 ^a^	<0.0001
	Central Nuclei	1.63 ± 0.41	4.87 ± 1.17 ^a^	<0.0001	1.63 ± 0.24	2.96 ± 0.49 ^a^	<0.0001
	Myonuclear Domain	1195.1 ± 208.3	909.6 ± 152.3 ^a^	0.0081	1160.6 ± 97.1	1019.1 ± 198.5	0.0998
	Nuclei/Sarcoplasm Area	0.010 ± 0.001	0.017 ± 0.001 ^a^	<0.0001	0.013 ± 0.001	0.012 ± 0.000	0.1082
	Total Connective Tissue	3.58 ± 0.56	6.83 ± 1.10 ^a^	<0.0001	4.31 ± 0.41	6.91 ± 1.38 ^a^	0.0008
	Epimysium	1.92 ± 0.30	3.35 ± 0.54 ^a^	<0.0001	2.32 ± 0.21	3.39 ± 0.68 ^a^	<0.0001
	Perimysium	0.64 ± 0.10	1.96 ± 0.31 ^a^	<0.0001	0.77 ± 0.07	1.98 ± 0.39 ^a^	<0.0001
	Endomysium	1.01 ± 0.15	1.51 ± 0.24 ^a^	0.0004	1.22 ± 0.11	1.52 ± 0.31	0.2689
	Collagen Type I	75.27 ± 3.49	66.32 ± 5.44 ^a^	0.0014	75.27 ± 6.36	78.52 ± 8.56	0.4048
	Collagen Type III	24.23 ± 3.49	33.67 ± 5.44 ^a^	0.0014	33.09 ± 5.86	21.47 ± 8.56 ^a^	0.0078
Fiber Types Profile	Proportion Fiber Type I	9.31 ± 1.49	5.40 ± 1.18 ^a^	<0.0001	86.05 ± 1.89	79.29 ± 1.35 ^a^	<0.0001
	Proportion Fiber Type IIA	41.68 ± 2.59	43.61 ± 3.08	0.1968	13.94 ± 1.89	20.71 ± 1.35	<0.0001
	Proportion Fiber Type IIB	49.00 ± 2.28	50.98 ± 2.16	0.0973	NA	NA	NA
	Cross-Sectional Area Type I	833.87 ± 26.92	779.70 ± 22.64 ^a^	0.0007	2923.4 ± 843.8	1669.2 ± 273.5 ^a^	<0.0001
	Cross-Sectional Area Type IIA	1158.1 ± 87.3	985.7 ± 131.6 ^a^	0.0093	4233.6 ± 685.6	3867.7 ± 463.9 ^a^	<0.0001
	Cross-Sectional Area Type IIB	2514.3 ± 149.2	2357.4 ± 292.4	0.2048	NA	NA	NA
Neuromuscular Junctions Structure	NMJ Cross-Sectional Area	151.38 ± 15.10	134.31 ± 7.75 ^a^	0.0167	161.64 ± 10.84	96.03 ± 6.55 ^a^	<0.0001
	NMJ Larger Diameter	25.71 ± 4.19	22.90 ± 1.53	0.1089	23.69 ± 0.94	16.17 ± 1.21 ^a^	<0.0001
	NMJ Smaller Diameter	10.03 ± 0.88	10.10 ± 1.23	0.8939	7.11 ± 0.40	5.58 ± 0.42 ^a^	<0.0001
	NMJ Diameter Ratio	2.56 ± 0.29	2.29 ± 0.22	0.0621	3.34 ± 0.30	2.90 ± 0.28 ^a^	0.0102
Antioxidant System	Superoxide Dismutase	6.61 ± 0.69	6.95 ± 0.70	0.3439	5.65 ± 0.25	5.64 ± 0.43	0.9409
	Catalase	403.37 ± 107.70	258.43 ± 84.11 ^a^	0.0101	204.45 ± 37.45	435.22 ± 28.13 ^a^	<0.0001
	Soluble Proteins	1.74 ± 0.06	1.81 ± 0.03 ^a^	0.0268	1.83 ± 0.07	1.91 ± 0.03 ^a^	0.0268
	Lipid Peroxides	27.60 ± 3.85	21.51 ± 2.95 ^a^	0.0405	13.44 ± 1.50	17.58 ± 1.37 ^a^	<0.0001
	Non-Protein Thiols	6.73 ± 0.44	5.42 ± 0.44 ^a^	<0.0001	7.37 ± 0.70	5.89 ± 0.45 ^a^	0.0003
	Total Cholinesterase’s	0.99 ± 0.06	0.85 ± 0.04 ^a^	0.0001	1.10 ± 0.12	0.84 ± 0.03 ^a^	0.0003

Legend: muscle weight (g/100 g), muscle length (mm), fiber density (number of fibers/mm^2^), cross-sectional area (µm^2^), larger and smaller diameters (µm), diameter ratio (larger diameter/smaller diameter), capillaries and nuclei (total number/total number fibers), central nuclei and proportion of fiber types (%), mionuclear domain (fiber cross-sectional area/total nuclei), nuclei/sarcoplasm area (nuclei cross-sectional area/fiber cross-sectional area). All analysis of connective tissue and collagen types (% pixels). Superoxide dismutase activity (U/mg protein); catalase activity (mM H_2_O_2_ consumed/min/mg protein); soluble proteins (mg/mL); lipid peroxides (nM hydroperoxides/mg protein); non-protein thiols concentration (nM thiols/mg protein); cholinesterase activity (nM acetylthiocholine hydrolyzed/min/mg protein). The letter a represents the difference between the MSG group when compared to the CTL group. All data: Student’s *t*-test.

## Data Availability

Zazula, Matheus; Zanardini de Andrade, Bárbara; Naliwaiko, Katya (2022), “MSG-obesity model—precocious inflammatory profile”, Mendeley Data, V3, doi:10.17632/td245zm5wg.3.

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
