# Peer review of "An Early and Sustained Inflammatory State Induces Muscle Changes and Establishes Obesogenic Characteristics in Wistar Rats Exposed to the MSG-Induced Obesity Model"

_ijms, 2023, doi:10.3390/ijms24054730_

Round 1

Reviewer 1 Report

The work is well constructed.

In the results section, in the tables is very difficult to follow the results. Perhaps the use of some figures may make the paper easier to follow. It is not cler where have yo measured the enzymes.

In the Material an methods section the method for catalese and the  other enzymes.... are explained as they were done in liver and I coud not see any value in the liver along the paper. This may be corrected.

In the discussion section you may remark their finding. Thy are writing about paper in the literature more than explaing deeper your findings

Author Response

1 – We appreciate this suggestion.

2 – We agree with the reviewer on this point and inform you that possible substitutions have been made. We are grateful for the suggestion, however, considering the number of variables evaluated in the study, replacing all tables with graphs would eventually produce a substantial increase in the size of the manuscript, especially when considering Table 3. we kept some data presented in the form of a table.

3 – After the note made by the reviewer, we revised the text and corrected it as requested.

4 – We accept the suggestion and revise the text of this section.

Reviewer 2 Report

Comments to the author:

The manuscript entitled “Long-term and precocious inflammatory systemic conditions contribute to the development of obesogenic characteristics and skeletal muscle alterations classics of the MSG-obesity model”. These data suggest that the showed that the animals showed reduced growth, increased adiposity, hyperinsulinemic and early pro-inflammatory conditions, chronic insulin resistance, reduced muscle mass and size, reduced muscle oxidative capacity, thickening of the extracellular matrix, reduced neuromuscular junctions and oxidative distress. 

Comments

1-This study fulfills their objective and it will open a new window for future research related to MSG

Author Response

1 – We are grateful for this comment.

Reviewer 3 Report

In this manuscript, authors claimed that early induction of MSG model led to increased adiposity, inflammation and changes in skeletal muscle which may contribute to later alternations on skeletal muscle and chronic obesity in rats. There are some interesting data but also still can be improved.

1. The observations of changes taken place at early age of rats only can only establish a possible association between all those complications with the chronic changes at later stage of development, it does not prove a necessary causal relationship.  

2. There are lots of grammar mistakes and misuse of words in the manuscript which made it difficult to understand and follow the logic flow, please revise. Such as “reduction of development” it was delayed but not reduced & “plasma TNFa secretion” should be “plasma level of TNFa” etc…

3. Please remove all P values in the results if it has already been presented in the figures or tables.

4. Fig 2B, it was an only trend but did not reach significance. Therefore, the statement in line 110-112 is inaccurate.

5. When evaluating insulin resistance, authors chose to conduct GTT. However, GTT measures more about glucose tolerance, for evaluation of insulin resistance or sensitivity, it is recommended to conduct ITT and calculate homeostatic model assessment index (HOMA-IR).

6. When discussing data presented in table 1 and 2, please follow the order of the table arrangement, do not jump back and forth between two tables.

7. In the discussion, authors cited lots of previous studies to support certain assumption, which seems to be too rush. More mechanistic studies should be conducted in order to reach such conclusions. Please revise.

8. Authors claimed increased insulin resistance in MSG animals with advanced adiposity and defected muscle development. Was this a systemic effects due to MSG or more related to development of IR at local level in tissues? If so, which tissue? Similar questions about inflammation, seems like MSG rats developed systemic inflammation based on circulating levels of some cytokines, how about local tissue level?

Author Response

1 – In fact, the simple observation of the results obtained in the two intervals may represent a fragile inference about this relationship. However, when we analyze the behaviour of the data using the Principal Coordinates analysis, it is identified that in both moments of investigation, the data present the same distribution between the experimental groups, intensifying the distancing of the components along the axes. This observation allows us to suggest that there may indeed be a causal relationship between the inflammatory findings, present since the PND14, and the later alterations found in the muscle.

2 – We inform you that the manuscript was submitted for revision regarding writing in English.

3 – We understand the reviewer's suggestion, but the presentation of significance values in the text, in addition to the tables, facilitates the understanding and distinction between results with or without significance. This way, the reader does not need to visit the tables to understand the results. In addition, statistically, this is the recommended way of presenting the results. We expect the understanding of the reviewer to maintain the values in the text.

4 – Regarding this note, we inform you that the comparison of day-to-day results does not allow inferring changes in food consumption. However, the use of Analysis of Variance of Repeated Measures, allows identifying that small daily changes, without punctual significance, add up over time, producing differences between the groups in the entire investigated time. Furthermore, this difference is also pointed out by the analysis of the area under the curve, which mathematically allows the integrating the data and confirms the results expressed over time.

5 – We agree that the ITT test represents a more accurate tool for the analysis of insulin resistance. However, it is known that the GTT test allows a safe inference about the insulin response of animals and that it is widely used due to its practicality. Given this, in the experimental design, the GTT was chosen and not the ITT. Considering that this analysis must be done in vivo, unfortunately, it is not possible to experiment in time to meet the deadlines for publication, as it would be necessary to obtain new animals specifically for this purpose. In addition, considering the results obtained from the insulin concentration during the GTT, we believe that they are sufficient data for the analysis of resistance without the obligation to perform the ITT.

Regarding the suggestion to include the HOMA-IR and QUICK mathematical tests, we inform you that the data obtained from this analysis were attached to the results.

6 – As requested, the text was corrected.

7 – We agree on the need for studies that elucidate the mechanisms involved in the model. However, we understand that the data obtained in this study were necessary to identify the characteristics caused by exposure to MSG, especially considering the ages evaluated. We believe that these data can help in understanding the effects of MSG for the identification of possible pathways and mechanisms of action to be investigated. We understand that starting evaluations through mechanistic studies represents a risky strategy that could lead to the identification of fragile and premature results.

8 – About RI, we clarify that the measurement was made in the plasma of animals, not being estimated in insulin-dependent tissues. Similarly, inflammatory parameters were also not evaluated in specific tissues. The measurement of circulating markers allows understanding of the systemic scenario, which may be participating in the effects of MSG. The investigation of markers in specific tissues can represent the context of a microenvironment, however, chronic degenerative diseases are characterized by several systemic effects that participate in the aggravation or characterization of a specific condition. Again, as the study hypothesis was based on understanding the installation and worsening of obesity in the model, we believe it is necessary to establish the systemic profile to propose personalized investigations.

Round 2

Reviewer 3 Report

Please add statistical information in all figure legends

Author Response

Thanks for the suggestions, we have inserted the information about the requested statistical tests. Again we revised the writing and requested points.